# STRUCTURING REASONING FOR COMPLEX RULES BEYOND FLAT REPRESENTATIONS

## ABSTRACT

Large language models (LLMs) face significant challenges when processing complex rule systems, as they typically treat interdependent rules as unstructured textual data rather than as logically organized frameworks. This limitation results in reasoning divergence, where models often overlook critical rule dependencies essential for accurate interpretation. Although existing approaches such as Chain-of-Thought (CoT) reasoning have shown promise, they lack systematic methodologies for structured rule processing and are particularly susceptible to error propagation through sequential reasoning chains. To address these limitations, we propose the Dynamic Adjudication Template (DAT), a novel framework inspired by expert human reasoning processes. DAT structures the inference mechanism into three methodical stages: *qualitative analysis, evidence gathering, and adjudication*. During the *qualitative analysis* phase, the model comprehensively evaluates the contextual landscape. The subsequent *evidence gathering* phase involves the targeted extraction of pertinent information based on predefined template elements ([placeholder]), followed by systematic verification against applicable rules. Finally, in the *adjudication* phase, the model synthesizes these validated components to formulate a comprehensive judgment. Empirical results demonstrate that DAT consistently outperforms conventional CoT approaches in complex rule-based tasks. Notably, DAT enables smaller language models to match, and in some cases exceed, the performance of significantly larger LLMs, highlighting its efficiency and effectiveness in managing intricate rule systems.

## 1 INTRODUCTION

Large language models (LLMs) (Zhao et al., 2025; Xiao et al., 2025) have exhibited remarkable capabilities across a wide range of natural language processing tasks (Vaswani et al., 2017; Brown et al., 2020), demonstrating significant potential in rule-intensive domains such as e-commerce content moderation (He et al., 2024), legal advisory services (Yue et al., 2024), and financial risk analysis (Wu et al., 2023b). These domains demand precise rule interpretation and rigorous implementation, as even minor errors in judgment can result in legal risks, financial losses, or serious damage to credibility. However, when processing complex instruction sets characterized by densely interdependent rules and nuanced semantic relationships (Ouyang et al., 2022), the performance of LLMs deteriorates significantly (Srivastava et al., 2022). For instance, when an advertisement must simultaneously convey the seller's promotional message and strictly comply with advertising laws, LLMs often struggle to navi-

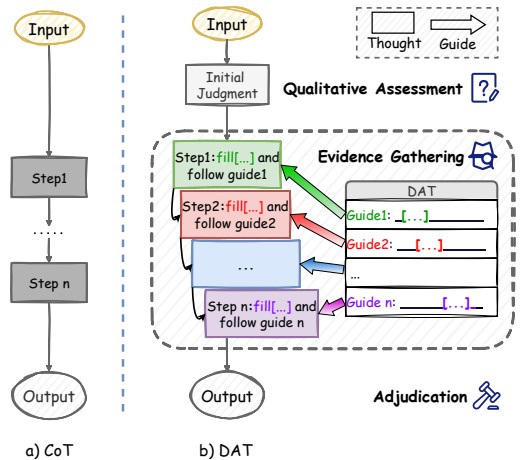

Figure 1: Illustration of different reasoning process. Dynamic Adjudication Template enables large language models to organizes the inference mechanism into a structured three-stage process.

gate the subtle interactions between overlapping rules, leading to inconsistent judgments (Xu et al., 2025). This limitation stems from two fundamental problems: (1) **Flat rule processing**, where models treat rule systems as unstructured text collections rather than hierarchically organized frameworks with explicit logical relationships; and (2) **Reasoning divergence**, where models are distracted by superficial features or misled by ambiguous cues (Ji et al., 2023), failing to identify the most relevant rules.

Existing reasoning approaches have attempted to address complex reasoning challenges but fall short in rule-intensive scenarios. Chain-of-Thought (CoT) (Wei et al., 2022) and its variants fail to ensure that each step strictly adheres to logic and facts. Errors in any intermediate step can propagate and collapse the coherence of the entire reasoning chain (Lyu et al., 2023; Yang et al., 2024b). Tree-based methods like Tree-of-Thought (ToT) (Yao et al., 2024) explore multiple paths but lack systematic rule verification. These approaches share a common limitation: they lack mechanisms for hierarchical rule organization, structured evidence verification, and systematic conflict resolution when multiple rules apply simultaneously.

This gap reveals a fundamental mismatch between current LLM reasoning approaches and the requirements of rule-intensive domains. Effective rule-based reasoning calls for a new framework that enables structured verification processes rather than relying on free-form exploration. To this end, we propose a novel method, the **Dynamic Adjudication Template (DAT)**, illustrated in **Figure 1.** This method emulates the cognitive approach of human experts (Bilalić, 2017; Liao & Varshney, 2021). Rather than beginning with detailed computation, experts first construct a high-level problem-solving framework and then identify critical points for focused analysis. Unlike static prompting techniques that rely on fixed templates, DAT introduces a dynamic, structured reasoning paradigm. It actively guides the model through a structured three-step process of *Qualitative Assessment, Evidence Gathering, and Adjudication.* The process begins with a high-level assessment of the problem. The model then focuses on key, complex, and error-prone decision points most relevant to the question, enabling targeted analysis. These targeted insights are finally synthesized into a logically sound and comprehensive adjudication. In our experiments on a rule-intensive e-commerce dataset, DAT improved the overall accuracy of Qwen-2.5-7B (Qwen et al., 2025) from 34.11% to 62.49%. It also outperformed larger CoT-equipped models such as Qwen-Max and Deepseek-R1 (DeepSeek-AI et al., 2025) on several complex rule-based subtasks within the dataset.

In summary, our contributions are threefold.

1. We propose the DAT approach, a human-inspired strategy that guides models through a structured three-step process: *Qualitative Assessment, Evidence Gathering, and Adjudication.* DAT transforms the reasoning paradigm from flat rule processing to hierarchical reasoning, and from uncontrolled error propagation to structured verification—enabling more coherent and rule-consistent judgments in complex tasks.

2. We introduce an automated pipeline for template generation, filtering, and selection. By replacing static prompts with adaptive templates, we enable flexible and context-aware reasoning across diverse rule scenarios. This design ensures both interpretability and adaptability, supporting controlled and effective reasoning in a wide range of rule-based tasks.

3. We report substantial gains on rule-intensive benchmarks. Our method enables small, efficient models to outperform much larger LLMs using Chain-of-Thought reasoning on complex rule-based tasks. This paves the way for high-performance applications in low-resource settings. Preliminary results also suggest promising generalization to vision-language models (VLMs) (Liang et al., 2024; Li et al., 2023), indicating a fruitful direction for future research.

## 2 RELATED WORK

### 2.1 COMPLEX RULE-BASED REASONING BENCHMARK

LLMs and VLMs have been widely adopted in e-commerce applications such as content moderation, product recommendation, and search (Jiang et al., 2024; Palen-Michel et al., 2024). Nevertheless, enabling these models to make accurate judgments under complex and dynamic platform rules remains a significant challenge. To evaluate model capabilities for such tasks, both academia and

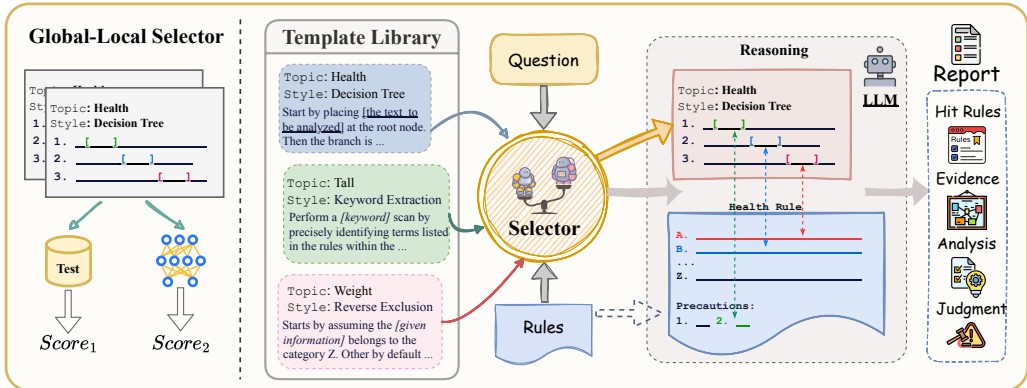

Figure 2: (Left) The architecture of the Global-Local Selector. (Right) When processing complex rule systems, LLM utilizes the selected DAT through Qualitative Analysis, Evidence Gathering, and Adjudication to generate a comprehensive judgment.

industry have developed specialized benchmarks. Among these, EVADE stands out as the first multimodal benchmark for evasive content detection in Chinese e-commerce. Derived from authentic advertising regulations and annotated by domain experts, this dataset comprises 2,833 textual samples and 13,961 images across six product categories. EVADE evaluates model performance via two distinct tasks: Single-Violation Judgment and All-in-One Judgment. By embedding semantically overlapping rules within instructions, it requires reasoning over complex, long-context, rule-dense scenarios, closely aligning with our focus on complex rule interpretation. Research using EVADE has revealed limitations in current models' capacity to comprehend and apply multi-layered rules, highlighting the necessity for our proposed approach.

## 2.2 Prompt-based Reasoning Methods with Large Language Models

To advance the reasoning capabilities of large language models, researchers have developed numerous prompt-based methodologies. Prominent among these are CoT prompting (Wei et al., 2022) and its derivatives, including Least-to-Most (Zhou et al., 2022a) and Decomposed Prompting (Khot et al., 2022), which decompose complex problems into sequential subtasks and have proven effective across diverse reasoning domains. However, the inherent structural fragility of linear reasoning approaches manifests as cascading error propagation during sequential processing stages, as demonstrated in Yang et al. (2024a). Emerging paradigms such as Tree-of-Thought and Graph-of-Thought (Besta et al., 2024) mitigate this issue through adaptive, non-linear heuristic search mechanisms. While these frameworks offer potential for improved reasoning fidelity, they introduce substantial computational costs and rely on handcrafted, domain-specific prompting strategies (Yang et al., 2024a). This rigid templating paradigm is particularly deficient for dynamic, evolving real-world environments. Within our problem domain, the governing rules exhibit both combinatorial complexity and temporal adaptation, rendering rule-category-specific manual prompt engineering impractical and misaligned with the dynamic nature of the challenge.

## 3 Methodology

In this section, we introduce our novel framework for rule-intensive applications. For a given query $Q$ and rules $\mathcal{R}$, the framework selects the optimal reasoning template $t^*$ from a predefined library $T_K$, which then guides the model in executing a structured reasoning process, as shown in **Figure 2.** To address the challenges of template generation, selection, and application, we propose a pipeline framework that consists of three primary components: 1) **Three-Stage Structured Reasoning.** This component serves as the core of our framework. It utilizes the selected template to execute a structured three-step reasoning process—*Qualitative Analysis, Evidence Gathering, and Adjudication*—to derive a logically grounded and rule-consistent result. This structured inference relies on high-quality, task-aligned templates, which are sourced and selected through the following two components. 2) **Dynamic Template Library Construction.** To support the reasoning process,

---

**Algorithm 1** Three-Stage Structured Reasoning Pipeline

---

**Input**: Question $q$, templates $T_K$, rules $\mathcal{R}$
**Parameter**: Template manager $\mathcal{M}$
**Output**: Judgment $\hat{J}$

1: **Step 1: Template Selection**
2: $t^* \leftarrow \mathcal{M}(q, T_K)$ // Select optimal template for the query.
3: **Step 2: Initial Judgment & Evidence Extraction**
4: $J_{\text{initial}} \leftarrow \text{QualitativeAnalysis}(q, t^*)$ // Form an initial, holistic judgment.
5: $\mathcal{P} \leftarrow \text{GetPlaceholders}(t^*)$ // Extract key placeholders from the template.
6: **for** each $p \in \mathcal{P}$ **do**
7: $\quad \mathcal{E}_p \leftarrow f_{\text{extract}}(p, q, \mathcal{R})$ // Extract evidence for each placeholder.
8: $\quad \mathcal{V}_p \leftarrow f_{\text{match}}(\mathcal{E}_p, \mathcal{R})$ // Match the evidence against rules.
9: **end for**
10: **Step 3: Evidence Integration**
11: Assemble evidence chain $\mathcal{C} \leftarrow \{\mathcal{V}_p\}_{p \in \mathcal{P}}$ // Assemble validated evidence.
12: $\hat{J} \leftarrow f_{\text{adjudicate}}(J_{\text{initial}}, \mathcal{C})$ // Adjudicate final judgment with evidence.
13: **Return** $\hat{J}$

---

this component systematically generates and validates a diverse repository of high-quality reasoning templates, which serve as the foundational knowledge base for the framework. 3) **Adaptive Template Selection.** Given a query, this component dynamically selects the optimal template from the library by evaluating both general performance and task-specific fitness. Detailed prompts for each component can be found in the **appendix D.2.**

### 3.1 THREE-STAGE STRUCTURED REASONING

To avoid reasoning divergence and error propagation in multi-step inference, we employ a structured, template-guided approach. As indicated in **Algorithm 1** and the right side of **Figure 2**, when encountering complex rules during reasoning, the model employs a dynamic template manager, which selects the appropriate template for the given task and question from a curated template knowledge base. The model then executes a structured three-step reasoning process: *Qualitative Analysis, Evidence Gathering, and Adjudication*. First, *Qualitative Analysis* guides the model to review all relevant information and form an initial, holistic judgment. This step prioritizes broad contextual understanding over premature attention to isolated details. Next, *Evidence Gathering* utilizes predefined [placeholders] within the selected template, representing **key reasoning checkpoints, complex decision nodes, and error-prone components,** to extract task-specific information from the question. The retrieved information is independently matched against relevant rules to verify consistency and correctness. Finally, *Adjudication* instructs the model to re-evaluate its initial judgment using the validated **evidence-rule chain**, producing a final decision through structured and rigorous logical reasoning. To support this reasoning process, we next describe how to construct a high-quality, dynamic template library.

### 3.2 TEMPLATE LIBRARY CONSTRUCTION

Our goal is to develop an efficient pipeline for generating data. This pipeline will build a diverse, high-quality, dynamically extensible library of reasoning templates. This repository will enable LLMs to adaptively select appropriate templates during their reasoning processes. The complete procedure is formalized in **Algorithm 2.**

**Template Generation and Expansion.** We introduce a systematic template generation pipeline using Gemini 2.5-Pro (Comanici et al., 2025). The process begins by creating 10 distinct seed templates ($T_0$) using only task context, without problem-specific details. These templates capture diverse analytical perspectives. To enhance coverage and robustness, we perform multi-stage augmentation through structured continuation (Zhou et al., 2022b; Madaan et al., 2023) and style transfer (Jin et al., 2022). Structured continuation begins by providing the model with the first several steps of each seed template—specifically, from step 1 to step $n-1$. The model is then instructed to complete the remaining steps in a consistent and logical manner, expanding $T_0$ into a structurally di-

---

**Algorithm 2** DAT Library Construction Pipeline

---

**Input**: Task context, full dataset $D$
**Parameters**:
Seed template count $m$, Prefix length $k$, Style Count $v$,
Sample ratio $r$ (default: 0.2), Performance threshold $\theta$
**Output**: Refined template set $T_K$, selector training data $S$
 1: **Step 1: Seed Template Generation**
 2: $T_0 \leftarrow$ GenerateSeeds(context, $m$) // Generate m initial seed templates.
 3: **Step 2: Template Expansion & Augmentation**
 4: $T_1 \leftarrow$ ExpandWithPrefix($T_0, k,$ LLM) // Expand seeds via structured continuation.
 5: $T_2 \leftarrow$ StyleTransfer($T_1, v$) // Apply v style variants to each template.
 6: **Step 3: Template Evaluation**
 7: $D_1 \leftarrow$ Sample($D, r$) // Randomly sample a subset $D_1$ for evaluation.
 8: $T_K \leftarrow \{t \in T_2 \mid \text{score}(t, D_1) \geq \theta\}$ // Filter templates based on performance score.
 9: **Step 4: Training Data Generation**
10: $S \leftarrow \{(\text{score}(t, d), t, d) \mid t \in T_K, d \in D_1\}$ // Construct training set S with scores for selector.
11: **Return** $T_K, S$ // Output the refined library and training data.

---

verse set $T_1$. We then apply style transfer to $T_1$ to introduce linguistic and syntactic diversity. By not specifying the exact styles, we harness the model's inherent understanding of language to produce a diverse and less biased set of augmented data. Specifically, the model is instructed to modify existing templates in $v$ distinct styles. This step modifies expressions while preserving the task relevance, yielding the final augmented set $T_2$ with improved stylistic diversity and task adaptability.

Template effectiveness is dynamically evaluated on a randomly sampled 20% subset of the data ($D_1 \subset D$), which serves as an approximate evaluation set for estimating generalizability. We assess each template in $T_2$ based on its ability to generate accurate and complete outputs for examples in $D_1$, and filter those that show reliable performance across varied examples in $D_1$, forming the refined library $T_K$. Performance records from applying $T_K$ to $D_1$ are then used to construct a training dataset $S$ for the template selector.

**Partitioning an Independent Dataset.** We construct an independent evaluation dataset $D_1$ by randomly sampling 20% of the complete dataset. This strategically sampled subset serves two key purposes in our pipeline. First, $D_1$ enables performance evaluation and filtering of the template library $T_2$. We measure critical metrics including error rates and task-specific accuracy across all templates in $T_2$. This assessment identifies high-performing templates that are optimally aligned with the task context, yielding a refined subset $T_K$. Second, $D_1$ generates training data for the template selector. We record the performance metrics of $T_K$'s predictions on $D_1$ to construct a supervised training dataset, which supports training the template selector discussed in Section Local Selector $S_2$. This enables context-aware, dynamic template selection across diverse task scenarios. With the template library constructed, we next introduce an adaptive selection mechanism to choose the most suitable template for each query.

### 3.3 ADAPTIVE TEMPLATE SELECTION

In rule-intensive scenarios, ensuring absolute controllability and determinism is critical. Directly using large-model-generated reasoning templates risks violating domain-specific constraints (Zhang et al., 2023). To address this, we propose a Global-Local Template Selector that draws from a curated repository of pre-validated templates (Lewis et al., 2020), ensuring constraint compliance and process reliability. As shown on the left of **Figure 2,** the selector retrieves and validates templates to align with rule-intensive constraints. Our dual-evaluation framework assesses both the **macro-level stability across tasks** and **adaptability to individual instances**. We identify optimal templates through a weighted integration of global and local performance metrics (Hastie et al., 2009), with min-max normalization applied to both scores:

$$s_{\text{final}}(T_i) = \lambda \cdot \frac{s_1(T_i) - \min_j s_1(T_j)}{\max_j s_1(T_j) - \min_j s_1(T_j)} + (1 - \lambda) \cdot \frac{s_2(T_i) - \min_j s_2(T_j)}{\max_j s_2(T_j) - \min_j s_2(T_j)}. \tag{1}$$

where $s_1$ denotes the global performance across tasks, $s_2$ the local fitness to the current query, and $\lambda \in [0, 1]$ a tunable hyperparameter balancing global stability and local fit.

**Global Selector $S_1$.** It evaluates templates based on their accuracy on $D_1$, reflecting general reliability across tasks. The global score $s_1$ is defined as:

$$s_1(T_i) = \text{Acc}(T_i, D_1). \tag{2}$$

Here, $T_i$ represents each template in the library, and $Acc(T_i, D_1)$ denotes the prediction accuracy of the template $T_i$ on the dataset $D_1$.

**Local Selector $S_2$.** To construct the training dataset, we pair reasoning templates that produce correct solutions with those that yield incorrect solutions for the same queries. Specifically, when template $T_A$ generates a correct result for a query and template $T_B$ generates an incorrect result, $T_A$ is considered superior to $T_B$ for that query. Using this method, we randomly sample 12,000 correct-incorrect template pairs from each of the six risk categories for training (Rafailov et al., 2023).

$$D_{\text{train}} = \{(T_A, T_B, Q) \mid Q \in D_1, \ T_A, T_B \in T_K, \ \mathbb{I}(R(T_A, Q)) = 1, \ \mathbb{I}(R(T_B, Q)) = 0\} \tag{3}$$

where $R(T, Q)$ denotes the result of applying template $T$ to query $Q$, and $(T_A, T_B)$ represents a superior–inferior template pair.

**DPO-based Training for $S_2$.** This training process is framed as a direct preference learning problem. The policy distribution $\pi_\theta(T_i|Q)$ models the probability of selecting template $T_i$ for a given question $Q$. This distribution depends on a learned preference score $r(Q, T_i)$, which quantifies how well $T_i$ suits $Q$. The parameter $\beta$ adjusts the strength of this preference. The score $r(Q, T_i)$ is derived from metrics such as output correctness, reasoning plausibility, or user feedback. Training uses pairs $(Q, T^+, T^-)$, where $T^+$ is preferred over $T^-$ for $Q$. The optimization objective $\mathcal{L}(\theta)$ maximizes the likelihood of selecting $T^+$ over $T^-$. Probabilities are normalized over the candidate template set $\mathcal{T}$. The template selection probability is modeled using the policy distribution $\pi_\theta(T_i|Q)$, which is defined as:

$$\pi_\theta(T_i|Q) = \frac{\exp(\beta \cdot r(Q, T_i))}{\sum_{T_j \in T_3} \exp(\beta \cdot r(Q, T_j))} \tag{4}$$

The optimization objective is trained by maximizing the log-likelihood function:

$$\mathcal{L}(\theta) = \mathbb{E}_{(Q, T^+, T^-)} \left[ \log \frac{\pi_\theta(T^+|Q)}{\pi_\theta(T^+|Q) + \pi_\theta(T^-|Q)} \right] \tag{5}$$

**Scoring with Local Selector $S_2$.** Under the preference learning framework, we aim for the local selector $S_2$ to score all templates in the template library at once, enabling it to identify the most suitable template for a given query. To achieve this, we pair the current query with each template in the template library and pass them into $S_2$, which calculates the probability of selecting a specific template for the given query. Templates with higher probabilities are assigned higher scores. For a template $T_i$ consisting of $m$ tokens, we compute the average negative log-likelihood (NLL) (Goodfellow et al., 2016) as follows:

$$s_2(T_i) = L_{\text{NLL}}(T_i) = -\frac{1}{m} \sum_{i=1}^{m} \log P(t_i \mid \text{context}(Q, T_i)) \tag{6}$$

where $\text{context}(Q, T_i)$ denotes the input formed by concatenating query $Q$ with template $T_i$. If the model assigns a high probability (close to 1) to each token $t_i$, the corresponding log term approaches 0, indicating a lower overall loss.

## 4 EXPERIMENTS

### 4.1 EXPERIMENTAL SETTINGS

**Implement details.** Template generation employs Gemini-2.5-Pro, leveraging its strengths in generating diverse and task-adaptive outputs. The optimal templates for subsequent training are selected through a refinement process based on Qwen2.5-7B-Instruct performance evaluated on the independent dataset $D_1$. For DPO, we apply low-rank adaptation (LoRA) to fine-tune the Qwen2.5-14B

base model efficiently. Training uses the Adam optimizer with an initial learning rate of $5 \times 10^{-6}$, and LoRA parameters: rank=16, alpha=32, dropout=0.05. The preference weight $\beta$ is set to 0.1. All experiments were executed using NVIDIA H20 GPUs.

**Complex Instruction Dataset.** Our experimental dataset is sourced from EVADE, containing 2,833 text and 13,961 image samples from real e-commerce platforms. The data focuses on six sensitive product categories: weight loss, diseases, height growth, body shaping, female care, and male care. Due to their direct impact on consumer welfare, these categories are governed by exceptionally detailed and strict policy rules derived from advertising laws. The logical complexity of these rules makes the dataset an ideal benchmark for evaluating a model's reasoning abilities. For your reference, I have included the prompt for one of the tasks, diseases, in the **appendix D.1.**

**Evaluation metrics.** Since the task format of EVADE requires the model to derive a final single-choice or multi-choice conclusion based on the multi-class rules in the prompt, our evaluation metrics align with this format. **Full Accuracy** $\text{Acc}_f$ requires the model's final prediction to exactly match the ground truth, while **Partial Accuracy** $\text{Acc}_p$ requires that the model's prediction has at least one overlap with the ground truth:

$$\text{Acc}_f = \frac{1}{N} \sum_{i=1}^{N} \mathbb{I}(C_i = G_i) \quad \Big| \quad \text{Acc}_p = \frac{1}{N} \sum_{i=1}^{N} \mathbb{I}(C_i \cap G_i \neq \varnothing)$$

where $N$ is the total number of samples, $C_i$ the predicted set, and $G_i$ the ground truth set for sample $i$. The item $\mathbb{I}(\cdot)$ denotes the indicator function, returning 1 if the condition is satisfied and 0 otherwise. Due to space constraints, we report Partial Accuracy ($\text{Acc}_p$) in the main text, while Full Accuracy results ($\text{Acc}_f$) are provided in **Table 4.**

Table 1: Performance Comparison of LLM Models on Partial Accuracy. Bold values represent the best performance across small parameter models, while underlined values indicate the best performance across large parameter models.

| Model | Method | Overall | Body | Women | Height | Men | Weight | Health |
|---|---|---|---|---|---|---|---|---|
| **Small Models (Baseline & Ours)** | | | | | | | | |
| ChatGLM3-6B | CoT | 25.39 | 43.83 | 42.60 | 24.60 | 30.84 | 21.19 | 4.68 |
| | Ours | 43.82 (↑18.43) | 67.28 | 62.72 | 39.73 | 48.28 | 29.94 | 23.26 |
| ChatGLM4-9B | CoT | 37.54 | 30.25 | 42.60 | 29.80 | 53.26 | 57.06 | 28.27 |
| | Ours | 59.57 (↑22.03) | 63.58 | **79.88** | 54.40 | 63.41 | 73.45 | 38.93 |
| InternLM3-8B | CoT | 45.54 | 39.51 | 37.28 | 52.82 | 61.88 | 51.69 | 39.58 |
| | Ours | 56.71 (↑11.17) | 58.64 | 64.50 | 59.82 | 63.79 | 65.25 | 40.23 |
| Qwen-2.5-7B | CoT | 34.11 | 24.69 | 37.87 | 41.08 | 53.83 | 38.14 | 29.56 |
| | Ours | **62.49 (↑28.38)** | **77.78** | 78.11 | **61.40** | 62.84 | 61.30 | **41.36** |
| Qwen-2.5-14B | CoT | 45.49 | 41.36 | 56.80 | 48.98 | 50.00 | 57.06 | 33.12 |
| | Ours | 58.16 (↑12.67) | 58.64 | 76.92 | 53.27 | **65.71** | **73.45** | 39.58 |
| **Large Models (Baseline Only)** | | | | | | | | |
| Qwen-2.5-72B | CoT | 51.16 | 62.35 | 61.54 | 40.86 | 57.09 | 58.76 | 33.28 |
| Qwen-max | CoT | 51.73 | 64.20 | 67.46 | 45.60 | 58.05 | 55.65 | 30.86 |
| GPT-4.1-0414 | CoT | 53.18 | 53.70 | 71.01 | 44.70 | 55.94 | 73.73 | 34.41 |
| Deepseek-R1 | CoT | 60.87 | 74.69 | 77.51 | 43.57 | 55.56 | 74.58 | 41.03 |

## 4.2 MAIN RESULTS

As shown in **Table 1,** the DAT framework significantly improves LLM performance on complex rule-based reasoning tasks. Two key findings emerge: (1) DAT consistently outperforms the standard CoT baseline across all models, and (2) it enables smaller models to match or exceed the performance of much larger, state-of-the-art systems.

**Superiority of DAT over CoT.** Performance comparisons show notable gains with DAT over CoT. For instance, Qwen-2.5-7B's "Body" score improved from 24.69 to 77.78, and ChatGLM4-9B's (GLM et al., 2024) "Women" score rose from 42.60 to 79.88. These consistent improve-

ments result from DAT's structured, rule-based reasoning, which curbs error propagation and avoids CoT's unguided steps. To ensure broad applicability, we evaluate DAT on diverse open-source LLMs—including Qwen-2.5, ChatGLM3/4, and InternLM3 (Team, 2023)—covering various architectures and sizes. We also benchmark four large-scale models (Qwen-2.5-72B, Qwen-Max, GPT-4.1, and DeepSeek-R1) under standard CoT to estimate upper-bound performance.

**Enabling Small Models to Rival Large Models.** Small DAT-enhanced models consistently outperform significantly larger CoT-based models. Qwen-2.5-7B with DAT surpassed Qwen-2.5-72B (62.35 vs. 53.70), GPT-4.1 (62.35 vs. 53.70), and Deepseek-R1 (62.35 vs. 69.14) on the "Body" metric. Similarly, for "Height", DAT-equipped Qwen-2.5-7B (61.40) exceeded all large CoT models. These results reveal that reasoning methodology—not model scale—is the primary constraint in rule-based reasoning. DAT overcomes this by enforcing hierarchical rule verification and preventing semantic confusion, enabling robust performance without massive parameter requirements.

### 4.3 ABLATIONS

**Impact of Template Selector.** To validate the effectiveness of the global selector $S_1$ and DPO-based selector S2, we conducted an ablation study comparing it against random template selection from the template library during inference. Results in Table 2 demonstrate that neither approach alone could achieve optimal template selection. To evaluate the effectiveness of the DPO-based template selector ($S_1$ and $S_2$), we conducted an ablation study against ran-

Table 2: Ablation results of template selector on Qwen-2.5-7B (Partial Accuracy).

| Strategy | Body | Women | Height | Men | Weight | Health |
|---|---|---|---|---|---|---|
| Random | 37.65 | 57.40 | 49.89 | 57.28 | 44.92 | 31.99 |
| $\lambda = 0$ | 58.02 | 59.17 | 59.59 | 62.45 | 44.92 | 30.05 |
| $\lambda = 0.3$ | 49.38 | 60.45 | 60.04 | 61.69 | 53.67 | 35.86 |
| $\lambda = 0.7$ | **77.78** | **78.11** | **61.40** | **62.84** | **61.30** | **41.36** |
| $\lambda = 1$ | 39.51 | 68.64 | 58.69 | 59.59 | 51.69 | 33.12 |

dom template selection and varied the weighting parameter $\lambda$ in the final score $s_{\text{final}} = \lambda s_1 + (1 - \lambda)s_2$ to assess the impact of global and local scores. As shown in **Table 2,** using only global ($\lambda = 1$) or local ($\lambda = 0$) scoring leads to suboptimal results. Performance improves when both are combined, with $\lambda = 0.7$—empirically tuned on representative settings—consistently achieving the best results. This highlights the complementary strengths of global stability and local adaptability, with the combined selector significantly outperforming the random baseline.

**The number of templates for scoring. Figure 3** shows multiple quantities of candidate templates. Our findings indicate that greater template diversity enhances the effectiveness of template selection for a given problem, thus facilitating enhanced task performance.

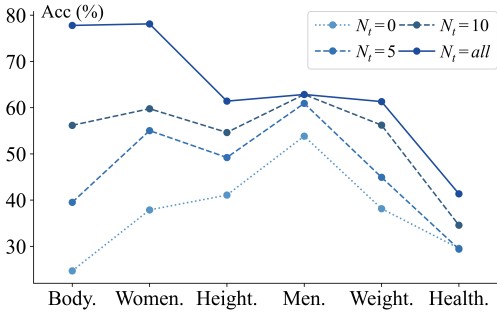

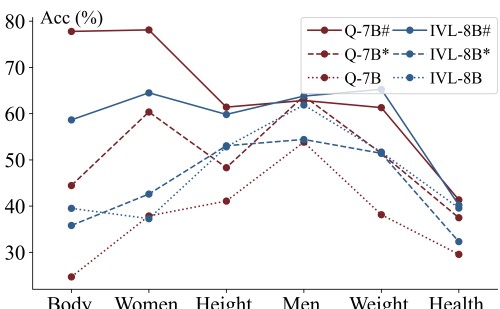

Figure 3: Performance Comparison of Different Numbers of Candidate Templates ($N_T$) on Qwen-2.5-7B.

Figure 4: Ablation of partial accuracy for Qwen-2.5-7B and InternLM3-8B. Baseline uses no enhancements; "*" adds evidence gathering; "#" adds evidence gathering and adjudication.

**The importance of the three-step reasoning process.** We further found that each stage of the structured three-step process, *Qualitative Assessment, Evidence Gathering, and Adjudication*, is essential. Relying solely on Qualitative Assessment leads to the same reasoning failures as the CoT baseline, as the model often forms premature judgments without deeper evidence-based verification. Crucially, the Adjudication stage is vital. As shown in **Figure 4,** models limited to evidence

extraction often overfocus on specific terms, leading to errors. The Adjudication step acts as a comprehensive review. This allows the model to reason objectively and make an informed final decision.

## 4.4 DISCUSSION

We extended DAT to high-capacity vision-language models (VLMs) and observed promising results. Without modifying the original pipeline, instructing VLMs to follow structured reasoning purely through textual templates proved effective on models with strong multimodal comprehension, such as Qwen-VL-Max (Bai et al., 2025), GPT-4.0 (OpenAI et al., 2024), and Gemini-2.5-Pro (**Table 3**). We focused on these large VLMs because they exhibit better understanding of complex textual instructions, which is essential for interpreting and executing structured tem-

Table 3: Performance Comparison of VLM Models on Partial Accuracy. Abbreviations: Claude = Claude 3.0, GPT = GPT-4o, Qwen-VL = Qwen-VL-Max.

| Task | Claude | | GPT-4o | | Qwen-VL-Max | |
|---|---|---|---|---|---|---|
| | CoT | Ours | CoT | Ours | CoT | Ours |
| Body | 67.97 | 74.45 | 67.29 | **82.45** | 65.42 | 80.75 |
| Women | 56.84 | 61.24 | 63.56 | 61.24 | 63.82 | **63.82** |
| Height | 65.14 | 73.68 | 62.60 | 71.65 | 70.73 | **75.61** |
| Men | 51.74 | 66.52 | 56.30 | **68.04** | 56.09 | 58.48 |
| Weight | 71.43 | **83.71** | 72.28 | 76.29 | 71.43 | 76.57 |
| Health | 43.93 | 51.86 | 44.90 | 46.76 | 43.93 | **50.32** |

plates. In contrast, smaller VLMs often show weaker performance in following detailed instructions (Liu et al., 2024; Wu et al., 2023a; Yang et al., 2023) and interpreting complex templates, likely due to their limited capacity. In our evaluation, this gap led to reduced effectiveness with DAT. We hypothesize that smaller VLMs lack sufficient understanding of and adherence to DAT, which limits their ability to deliver optimal performance. This highlights a potential avenue for improvement: by tailoring the design of templates for vision-language tasks, it may be possible to enhance the performance of smaller VLMs in future work.

## 4.5 CASE STUDY

**Figure 5** compares CoT and DAT on a height-related regulation case. CoT identifies the violation "grow taller" but overlooks the exemption for minors. In contrast, DAT follows a structured template to extract evidence, apply rules, and identify exemptions—ultimately concluding that the case falls under Rule Z. This demonstrates DAT's ability to perform rule-consistent, multi-step reasoning. More examples can be found in the **appendix C.**

## 5 CONCLUSION

In this paper, we address the limitations of LLMs in interpreting complex rule-based systems, especially in high-stakes areas like law and finance. To tackle this, we propose the DAT, a novel framework inspired by human cognitive processes that guides the model through a structured three-stage reasoning process: Qualitative Assessment, Evidence Gathering, and Adjudication.

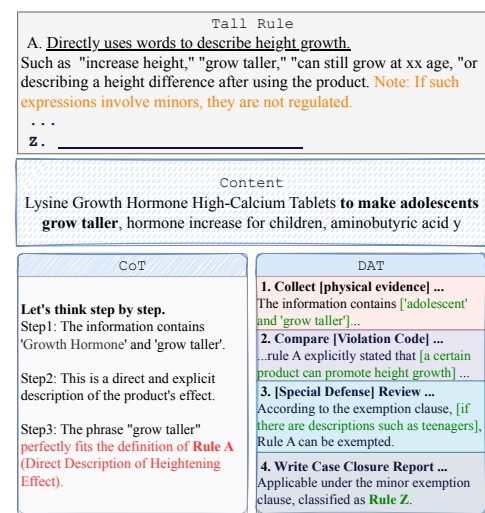

Figure 5: Case study contrasting DAT's structured reasoning flow with CoT's flat response, showing how DAT aligns rules and evidence to reach a consistent decision.

Unlike prior methods using unstructured reasoning or static prompts, DAT extracts targeted evidence via dynamic placeholders and performs independent rule matching for integrated logic synthesis. An automated pipeline handles template generation, filtering, and task-specific selection, ensuring adaptability and precision across diverse rule systems. Experiments on complex rule-based tasks show that DAT significantly enhances the accuracy and reasoning of smaller models, outperforming standard CoT and even surpassing state-of-the-art LLMs on certain tasks. These results highlight DAT's effectiveness in resource-constrained settings and its broader applicability to VLMs.

ETHICS STATEMENT

During the development of the DAT, we strictly adhered to ethical guidelines and legal regulations, ensuring fairness, transparency, inclusivity, and respect for all stakeholders. We emphasized the importance of protecting privacy and intellectual property, reaffirming our commitment to responsible and lawful data management. The e-commerce dataset used in our experiments was handled with care to mitigate ethical risks. All data was anonymized to protect privacy, and we acknowledge that it may contain inherent biases reflecting the time and context of its collection. These biases do not represent the views of the authors. We encourage future users of our method to be vigilant about evaluating and mitigating fairness and bias concerns in their specific applications. The EVADE-Bench may contain expressions and visual materials influenced by objective factors such as the time of collection, cultural context, and business scenarios.

REPRODUCIBILITY STATEMENT

This work prioritizes reproducibility through comprehensive documentation of all experimental components. In the appendix, we provide detailed experimental specifications including random seed settings, API source information, and inference temperature parameters. For training procedures, we fully disclose the GPU configurations, number of epochs, learning rates, and batch sizes. Additionally, we present two rigorous pseudocode algorithms that explicitly describe our library construction process and inference pipeline. All code, datasets, and model checkpoints will be made publicly available upon publication. Our implementation relies primarily on widely-available open-source libraries and frameworks. The detailed parameter settings and step-by-step procedures in our appendix should enable other researchers to replicate our results with minimal ambiguity. For experiments involving commercial API calls, we specify the exact API source. All reported results are computed with fixed seeds to ensure statistical reliability.

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

## A    STATEMENT ON THE USE OF LLMS

We employed Large Language Models (LLMs) as an auxiliary tool in preparing this manuscript. The LLM's function was strictly confined to assisting with two tasks: polishing the English prose and generating simple Python code snippets. All core research components—including the conceptualization, experimental design, data analysis, and the formulation of conclusions—were conducted exclusively by the human authors through collaborative discussion. Therefore, the LLM served as a support tool and did not contribute to the intellectual work of this study.

# B DETAILED PERFORMANCE OF ALL MODELS WITH DAT

Due to space constraints in the main paper, we only reported the partial accuracy metric on EVADE-Bench. Here, we present the performance of each model across all metrics within the six categories.

| Model | Method | Body | Women | Height | Men | Weight | Health |
|---|---|---|---|---|---|---|---|
| **Small Models (Baseline & Ours)** | | | | | | | |
| ChatGLM3-6B | Baseline | 43.8 / 21.6 | 42.6 / 7.1 | 24.6 / 9.0 | 30.8 / 14.9 | 21.2 / 5.1 | 4.7 / 2.8 |
| | Ours | 67.3 / 54.9 | 62.7 / 46.2 | 39.7 / 35.4 | 48.3 / 44.8 | 29.9 / 7.1 | 23.3 / 20.8 |
| | Δ | +23.5 / +33.3 | +20.1 / +39.1 | +15.1 / +26.4 | +17.4 / +29.9 | +8.8 / +2.0 | +18.6 / +18.1 |
| ChatGLM4-9B | Baseline | 30.2 / 29.6 | 42.6 / 37.9 | 29.8 / 14.2 | 53.3 / 45.0 | 57.1 / 20.9 | 28.3 / 20.8 |
| | Ours | 63.6 / 61.1 | **79.9** / 63.9 | 54.4 / 42.7 | 63.4 / 56.1 | 73.4 / 20.6 | 38.9 / 34.7 |
| | Δ | +33.3 / +31.5 | +37.3 / +26.0 | +24.6 / +28.4 | +10.1 / +11.1 | +16.4 / -0.3 | +10.7 / +13.9 |
| InternLM3-8B | Baseline | 39.5 / 30.2 | 37.3 / 20.7 | 52.8 / 26.6 | 61.9 / 45.2 | 51.7 / 9.6 | 39.6 / 23.3 |
| | Ours | 58.6 / 56.2 | 64.5 / 59.8 | 59.8 / 51.2 | 63.8 / 57.1 | 65.2 / 17.0 | 40.2 / 33.9 |
| | Δ | +19.1 / +25.9 | +27.2 / +39.1 | +7.0 / +24.6 | +1.9 / +11.9 | +13.6 / +7.4 | +0.6 / +10.7 |
| Qwen-2.5-7B | Baseline | 24.7 / 22.2 | 37.9 / 34.3 | 41.1 / 24.4 | 53.8 / 45.2 | 38.1 / 11.3 | 29.6 / 20.5 |
| | Ours | **77.8** / **74.1** | 78.1 / **71.0** | **61.4** / **48.8** | 62.8 / 56.9 | 61.3 / 16.7 | **41.4** / **32.0** |
| | Δ | +53.1 / +51.9 | +40.2 / +36.7 | +20.3 / +24.4 | +9.0 / +11.7 | +23.2 / +5.4 | +11.8 / +11.5 |
| Qwen-2.5-14B | Baseline | 41.4 / 38.9 | 56.8 / 47.3 | 49.0 / 28.2 | 50.0 / 36.8 | 57.1 / 20.9 | 33.1 / 19.7 |
| | Ours | 58.6 / 54.3 | 76.9 / 60.9 | 53.3 / 39.0 | **65.7** / **55.9** | **73.4** / 26.8 | 39.6 / 28.4 |
| | Δ | +17.3 / +15.4 | +20.1 / +13.6 | +4.3 / +10.8 | +15.7 / +19.2 | +16.4 / +5.9 | +6.5 / +8.7 |
| **Large Models (Baseline Only)** | | | | | | | |
| Qwen-2.5-72B | Baseline | 62.4 / 53.7 | 61.5 / 32.5 | 40.9 / 24.8 | 57.1 / 42.0 | 58.8 / 17.2 | 33.3 / 17.1 |
| | Ours | 84.6 / 79.6 | 75.7 / 70.4 | 43.3 / 27.8 | 68.0 / 57.8 | 73.4 / 25.1 | 43.5 / 22.3 |
| | Δ | +22.2 / +25.9 | +14.2 / +37.9 | +2.5 / +2.9 | +10.9 / +15.9 | +14.7 / +7.9 | +10.2 / +5.2 |
| Qwen-max | Baseline | 64.2 / 57.4 | 67.5 / 52.7 | 45.6 / 28.0 | 58.0 / 45.6 | 55.6 / 20.3 | 30.9 / 19.1 |
| | Ours | 81.5 / 78.4 | 83.4 / 78.7 | 54.6 / 34.5 | 64.0 / 54.0 | 74.6 / 24.9 | 38.4 / 25.8 |
| | Δ | +17.3 / +21.0 | +16.0 / +26.0 | +9.0 / +6.6 | +5.9 / +8.4 | +18.9 / +4.5 | +7.6 / +6.8 |
| GPT-4.1-0414 | Baseline | 53.7 / 46.9 | 71.0 / 56.2 | 44.7 / 23.0 | 55.9 / 42.2 | 73.7 / 24.9 | 34.4 / 18.4 |
| | Ours | 72.2 / 63.0 | 76.3 / 71.0 | 54.2 / 36.3 | 63.8 / 50.6 | 78.2 / 24.9 | 38.6 / 25.8 |
| | Δ | +18.5 / +16.0 | +5.3 / +14.8 | +9.5 / +13.3 | +7.8 / +8.4 | +4.5 / +0.0 | +4.2 / +7.4 |
| Deepseek-R1 | Baseline | 74.7 / 69.1 | 77.5 / 41.4 | 43.6 / 16.9 | 55.6 / 37.7 | 74.6 / 26.6 | 41.0 / 18.1 |
| | Ours | 82.7 / 79.6 | 81.1 / 58.0 | 49.4 / 21.2 | 67.4 / 54.4 | 83.3 / 22.9 | 45.2 / 17.3 |
| | Δ | +8.0 / +10.5 | +3.6 / +16.6 | +5.9 / +4.3 | +11.9 / +16.7 | +8.8 / -3.7 | +4.2 / -0.8 |

Table 4: Performance Comparison of LLM Models on Various Metrics. Bold values represent the best performance across small parameter models, while underlined values indicate the best performance across large parameter models.

| Model | Method | Body | Women | Height | Men | Weight | Health |
|---|---|---|---|---|---|---|---|
| **VLM Models** | | | | | | | |
| Claude3-7 | Baseline | 68.0 / 30.0 | 56.8 / 38.2 | 65.1 / 21.0 | 51.7 / 33.3 | 71.4 / 8.3 | 43.9 / 23.2 |
| | Ours | 74.4 / 33.6 | 61.2 / 46.5 | 73.7 / 32.0 | 66.5 / 46.7 | 83.7 / 19.1 | 51.9 / 22.4 |
| | Δ | +6.5 / +3.6 | +4.4 / +8.3 | +8.5 / +11.0 | +14.8 / +13.5 | +12.3 / +10.8 | +7.9 / -0.8 |
| GPT-4o | Baseline | 67.3 / 29.5 | 63.6 / 47.0 | 62.6 / 24.9 | 56.3 / 36.1 | 72.3 / 9.7 | 44.9 / 31.8 |
| | Ours | **82.4 / 40.0** | 61.2 / **48.3** | 71.6 / **36.9** | **68.0 / 50.2** | 76.3 / 10.3 | 46.8 / **34.9** |
| | Δ | +15.2 / +10.6 | -2.3 / +1.3 | +9.0 / +12.0 | +11.7 / +14.1 | +4.0 / +0.6 | +1.9 / +3.1 |
| Qwen-VL-Max | Baseline | 65.4 / 25.9 | 63.8 / 40.8 | 70.7 / 31.9 | 56.1 / 34.1 | 76.6 / 11.7 | 45.8 / 22.0 |
| | Ours | 80.8 / 32.9 | **64.6** / 40.3 | **75.6** / 35.1 | 58.5 / 40.0 | **86.6 / 13.4** | **50.3** / 31.1 |
| | Δ | +15.3 / +7.0 | +0.8 / -0.5 | +4.9 / +3.2 | +2.4 / +5.9 | +10.0 / +1.7 | +4.5 / +9.1 |

Table 5: Performance Comparison of VLM Models on Various Metrics. Bold values represent the best performance across all VLM models for a given metric.

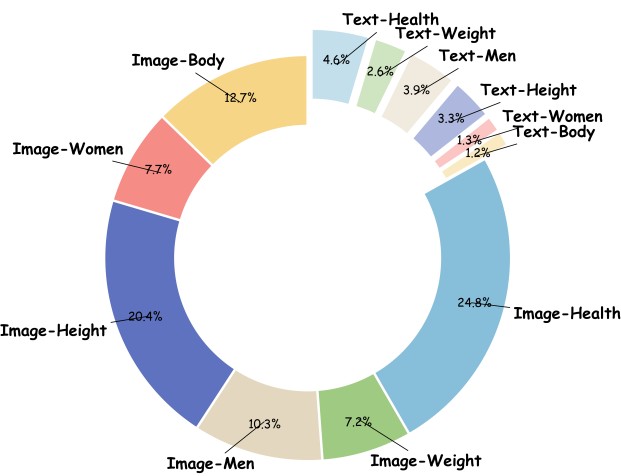

| Data Type | Text Count | Image Count |
|---|---|---|
| Body Shaping | 202 | 2,134 |
| Women's Health | 211 | 1,295 |
| Height Growth | 553 | 3,424 |
| Men's Health | 652 | 1,738 |
| Weight Loss | 442 | 1,203 |
| Health Supplement | 773 | 4,167 |
| Overall | 2,833 | 13,961 |

Table 6: The data distribution of EVADE-Bench for each violation category.

# C  SOME CASE OF EVADE-BENCH

Due to space limitations, here we mainly list several text&image samples of EVADE-Bench and common erroneous conclusions of the model.

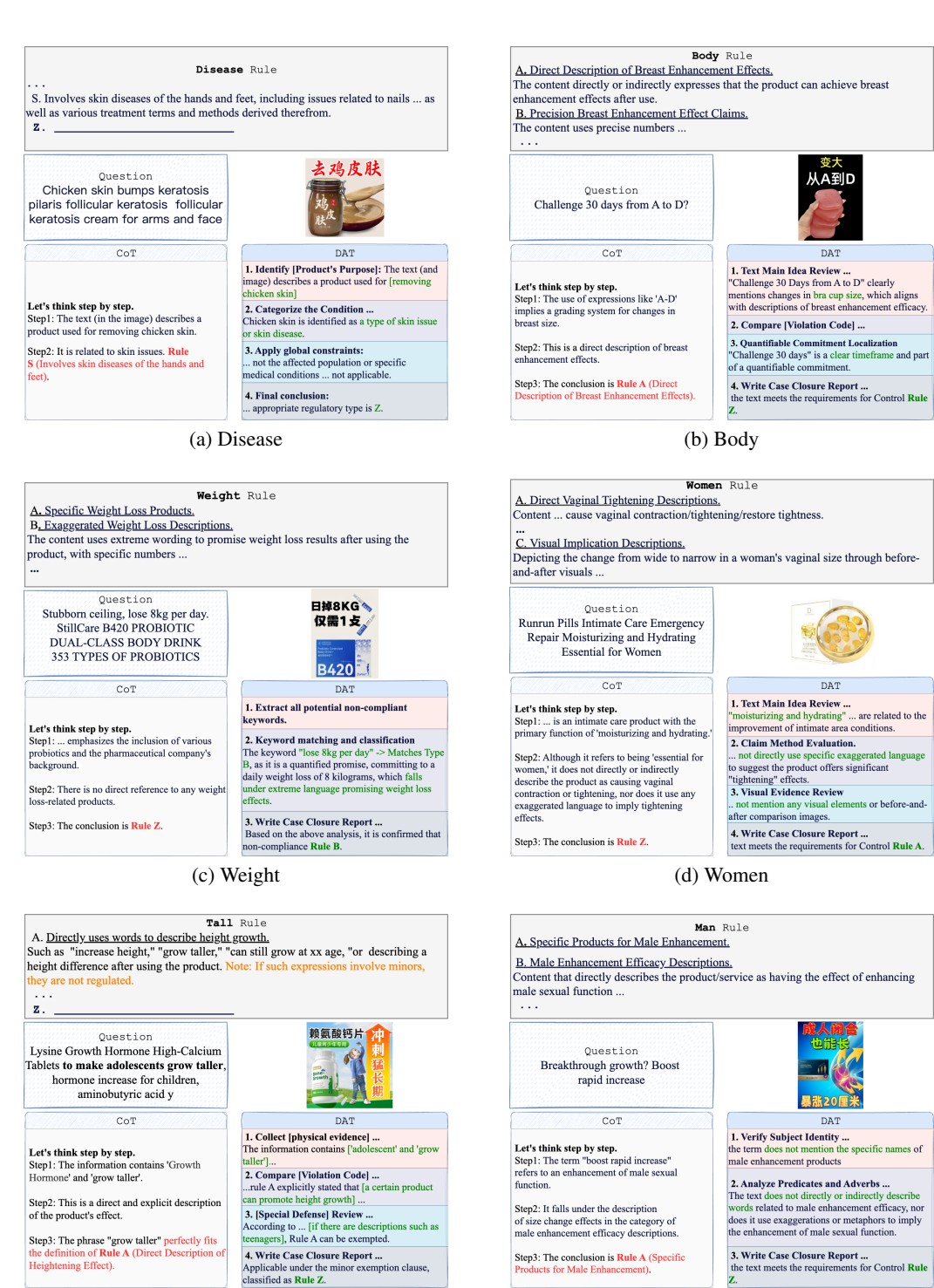

Figure 6: Examples showcasing the effectiveness of DAT across six tasks.

## D PROMPT

### D.1 RULE PROMPT EXAMPLES

The following are the rule definitions for the "health" task from the EVADE dataset. The domain is characterized by numerous violation types and challenging logical judgments.

---

**The prompt of Health products in Single violation - ZH**

# 任务描述
请你根据给定信息和关键词，仔细判断对应的管控类型。给出最终选项（一个到多个）。都不相关返回 Z.其他。

# 管控类型
- A.甲乙丙类传染病
- B.心脏类疾病：涉及心脏相关的疾病问题（包含心梗、心绞、冠心、心率、心力、血栓、心慌、心悸、胸闷等），同时还包含血管硬化、血管堵塞、血粘稠等血管血液的疾病问题和因此衍生的各种治疗用语与治疗手段。
- C.严重神经系统疾病：仅涉及神经系统相关的脑卒中、老年痴呆、中风、脑膜炎、脑梗、脑萎缩这几种严重疾病和因此衍生的各种治疗用语与治疗手段。
- D.严重内分泌疾病：仅涉及糖尿病酮症酸中毒、甲亢、甲减、桥本甲状腺等严重疾病和因此衍生的各种治疗用语与治疗手段。
- E.严重创伤和烧伤：仅涉及多发性创伤、大面积烧伤和因此衍生的各种治疗用语与治疗手段。
- F.生育类疾病：仅涉及生育的疾病问题（如不孕、不育、多囊卵巢综合征、精子畸形等）和因此衍生的各种治疗用语与治疗手段。
- G.癌症肿瘤类疾病：涉及人体各个部位的瘤、肿瘤、癌症、癌变、癌扩散等疾病和因此衍生的治疗用语与治疗手段。
- H.严重精神类疾病：仅涉及重度抑郁症、精神分裂等严重疾病和因此衍生的各种治疗用语与治疗手段。
- I.结石结节类疾病+通用治疗效果：涉及人体结石或结节类疾病问题的同时，还需要出现通用治疗效果词。比如仅描述"肾结石"不属于该分类，但既描述"肾结石"又提及"清除"的字眼，就属于该分类。
- J.痔疮类疾病+通用治疗效果：涉及人体痔疮类疾病问题的同时，还需要出现通用治疗效果词。比如仅描述"小肉球"不属于该分类，但既描述"小肉球"又提及"根治"的字眼，就属于该分类。
- K.生殖器官疾病+通用治疗效果：涉及男性生殖器官疾病或女性生殖器官疾病的同时，还需要出现通用治疗效果词。比如仅描述"尿道"不属于该分类，但既描述"尿道"又提及"突破"的字眼，就属于该分类。它与上面生育类疾病的区别在于，生殖器官疾病不包含生育问题。
- L.幽门螺旋杆菌+通用治疗效果：涉及幽门螺旋杆菌疾病问题的同时，还需要出现通用治疗效果词。比如仅描述"幽"不属于该分类，但既描述"幽"又提及"清"或"去"等字眼，就属于该分类。
- M.五官类疾病：涉及牙齿、耳朵、鼻子、头发、眼睛等人体五官的疾病问题和因此衍生的各种治疗用语与治疗手段。
- N.三高类疾病：涉及血压、血脂、血糖等疾病问题和因此衍生的各种治疗用语与治疗手段。
- O:肺部呼吸类疾病：涉及呼吸窘迫综合症ARDS、肺炎、肺病COPD、以及所有人体肺部器官的疾病问题和因此衍生的各种治疗用语与治疗手段。
- P.胃部消化类疾病：涉及急性胰腺炎、消化道出血、以及所有人体胃部器官的疾病问题和因此衍生的各种治疗手段。
- Q.肝部类疾病：涉及肝衰竭、肝硬化、以及所有肝部器官的疾病问题和因此衍生的各种治疗用语与治疗手段。
- R.关节类疾病：涉及关节疼痛、增生、麻木、畸形、肿痛、尿酸等体现在关节上的疾病和因此衍生的各种治疗用语与治疗手段。
- S.手脚皮肤类疾病：涉及指甲、灰甲、空甲、脚气、手脚糜烂、水泡、藓类、瘊疣类、湿疹类、抑菌止痒类、狐臭类疾病问题和因此衍生的各种治疗用语与治疗手段。
- T.轻度疾病：涉及感冒、鼻炎、焦虑症、失眠、睡不着、深睡等一系列睡眠类疾病问题和衍生的治疗用语与治疗手段。
- U.医用级描述：包含"医用级"、"医疗级"、"治愈"、"疗效"、"神药"、"奇效"、"专用药"等含义的内容，比如"瘙痒特效药"、"专攻哮喘"、"血糖百试百灵"等宣传描述。
- V.代替医疗类：包含"取代"、"比*好用"、"告别"等表达，并提及以下药品名称(他汀、双胍、4联/四联、甲钴胺、地屈孕酮、非布司他、二甲、黄体酮、所有的药和所有的疫苗)或医疗手段(手术、开刀、打针、注射、拉皮、拍CT、戴助听、做耳蜗、复查、胃镜、医美、所有的针（如水光针、美白针))，比如仅描述"疫苗"不属于该分类，但既描述"疫苗"又提及"吊打"的字眼，就属于该分类。
- W.代替医美类：包含"填充"、"膨胀"、"丰满"、"挺起"、"缩小"等形状变化的表达，同时需要提及以下身体部位(胸部、鼻子、眼睛)，比如仅描述"胸"不属于该分类，但既描述"挺"又提及"胸"的字眼，就属于该分类。
- X.无副作用类描述：包含"无副"、"无副作用"、"无风险"、"零风险"等表达商品无副作用和风险的内容，或者包含"无药"、"无药物残留"、"无毒"、"0化学"、"0农药"等表达商品无化学或农药添加剂的内容，或者包含"安全"、"安全性高"、"副作用小"等表达商品使用风险低的内容。
- Z. 其他：当给定信息与上述选项均无明显关联时选择此项。

# 注意事项
1. 谨慎判断:若无法正确识别，可能会加重患者病情或导致其他严重后果，请务必谨慎、细致地判断。
2. 权威引用:若文本明确引用百度百科、搜狗百科、维基百科、书籍内容或书名等权威信息来源，判断为"不相关"返回Z.其他。
3. 科普内容:若文本仅为科普（如书籍、电影等），且未出现具体商品名称，判断为"不相关"返回Z.其他。
4. 患病人群描述:若文本仅描述患病人群（如描述艾滋病人、心脏病患者的书籍或活动），判断为"不相关"返回Z.其他。
5. 通用治疗效果:文本可能会出现通用治疗效果相关的词汇，这些效果表现类词汇主要用于描述事物在功能或状态上的变化，突出其带来的积极效果。它们可以分为多个类别:上升类词汇如"拯救"、"改善"、"提高"、"增强"、"加强"、"提升"等，强调对某种能力或状态的提升和优化；下降类词汇如"抗"、"抑制"、"降"、"镇"、"缓解"、"控制"等，侧重于减少、不良因素的抑制或控制；消除类词汇如"告别"、"根治"、"消失"、"彻底"、"治愈"、"摆脱"、"解决"等，强调对问题或障碍的彻底消除和根除；用量类词汇如"仅需"、"只要"、"只需"等，突出使用的简便性和低成本。此外，其他类词汇如"有效率"、"治愈率"等，强调效果的高效性和成功率。这些词汇的共同特征在于通过具体的动词和形容词，传达出显著的效果和优势。

---

---

**The prompt of Health products in Single violation - EN**

# Task Description
Based on the given information and keywords, please carefully determine the corresponding control type. Provide the final option(s) (one or more). If none are relevant, return Z. Other.

# Control Types
- A. Class A, B, and C infectious diseases
- B. Cardiac diseases: Involves heart-related conditions (including myocardial infarction, angina, coronary heart disease, heart rate issues, heart failure, thrombosis, heart palpitations, chest tightness, etc.), as well as vascular and blood diseases like arteriosclerosis, vascular blockage, and high blood viscosity, and the various treatment terms and methods derived from them.
- C. Severe nervous system diseases: Only involves these severe nervous system diseases: stroke, Alzheimer's disease, apoplexy, meningitis, cerebral infarction, and cerebral atrophy, and the various treatment terms and methods derived from them.
- D. Severe endocrine diseases: Only involves severe diseases such as diabetic ketoacidosis, hyperthyroidism, hypothyroidism, and Hashimoto's thyroiditis, and the various treatment terms and methods derived from them.
- E. Severe trauma and burns: Only involves multiple trauma and extensive burns, and the various treatment terms and methods derived from them.
- F. Fertility-related diseases: Only involves fertility-related disease issues (such as infertility, sterility, polycystic ovary syndrome, sperm deformity, etc.) and the various treatment terms and methods derived from them.
- G. Cancer and tumor-related diseases: Involves tumors, neoplasms, cancers, carcinogenesis, cancer metastasis, etc., in various parts of the human body, and the derived treatment terms and methods.
- H. Severe mental illnesses: Only involves severe illnesses such as major depressive disorder and schizophrenia, and the derived treatment terms and methods.
- I. Calculus and nodule diseases + General therapeutic effects: Involves human calculus or nodule diseases and must also include general therapeutic effect terms. For example, describing only "kidney stones" does not belong to this category, but describing both "kidney stones" and mentioning a word like "clear" belongs to this category.
- J. Hemorrhoid-related diseases + General therapeutic effects: Involves human hemorrhoid-related diseases and must also include general therapeutic effect terms. For example, describing only a "small fleshy lump" does not belong to this category, but describing both a "small fleshy lump" and mentioning a word like "eradicate" belongs to this category.
- K. Genital organ diseases + General therapeutic effects: Involves male or female genital organ diseases and must also include general therapeutic effect terms. For example, describing only the "urethra" does not belong to this category, but describing both the "urethra" and mentioning a word like "breakthrough" belongs to this category. It differs from the fertility diseases above in that genital organ diseases do not include fertility issues.
- L. Helicobacter pylori + General therapeutic effects: Involves Helicobacter pylori disease issues and must also include general therapeutic effect terms. For example, describing only "H. pylori" does not belong to this category, but describing "H. pylori" and mentioning words like "clear" or "remove" belongs to this category.
- M. Diseases of the five sense organs: Involves diseases of the teeth, ears, nose, hair, and eyes, and the derived treatment terms and methods.
- N. "Three Highs" diseases: Involves diseases related to high blood pressure, high blood lipids, and high blood sugar, and the derived treatment terms and methods.
- O: Pulmonary and respiratory diseases: Involves Acute Respiratory Distress Syndrome (ARDS), pneumonia, Chronic Obstructive Pulmonary Disease (COPD), and all diseases of the human lungs, and the derived treatment terms and methods.
- P. Gastric and digestive diseases: Involves acute pancreatitis, digestive tract hemorrhage, and all diseases of the human stomach, and the derived treatment terms and methods.
- Q. Hepatic (liver) diseases: Involves liver failure, cirrhosis, and all diseases of the human liver, and the derived treatment terms and methods.
- R. Joint-related diseases: Involves diseases manifested in the joints such as joint pain, hyperplasia, numbness, deformity, swelling, and uric acid issues, and the derived treatment terms and methods.
- S. Hand, foot, and skin diseases: Involves issues with nails, onychomycosis, koilonychia, athlete's foot, hand/foot erosion, blisters, tinea, warts, eczema, antibacterial/anti-itch conditions, and body odor, and the derived treatment terms and methods.
- T. Mild illnesses: Involves a series of sleep-related issues like the common cold, rhinitis, anxiety disorders, insomnia, inability to sleep, and deep sleep problems, and the derived treatment terms and methods.
- U. Medical-grade descriptions: Includes content with meanings like "medical-grade," "therapeutic efficacy," "miracle drug," "miraculous effect," "specialized medicine," etc. For example, promotional descriptions like "special effect drug for itching," "specializing in asthma," "infallibly effective for blood sugar."
- V. Medical replacement claims: Includes expressions like "replace," "better than *," "say goodbye to," and mentions the following drug names (statins, biguanides, quadruple therapy, metformin, dydrogesterone, febuxostat, etc., all drugs and all vaccines) or medical procedures (surgery, injections, facelifts, CT scans, hearing aids, cochlear implants, follow-up checks, gastroscopy, cosmetic procedures, all types of cosmetic injections, etc.). For example, merely describing "vaccine" does not fall into this category, but describing "vaccine" while also mentioning a word like "outperforms" does.
- W. Cosmetic procedure replacement claims: Includes expressions of shape change like "fill," "plump," "fullen," "lift up," "shrink," while also mentioning the following body parts (breasts, nose, eyes). For example, merely describing "breasts" does not fall into this category, but mentioning both "lift" and "breasts" does.
- X. No side-effect claims: Includes content expressing that a product has no side effects or risks, such as "no side effects," "no risk," "zero risk." Or content expressing the product has no chemical or pesticide additives, such as "drug-free," "no drug residue," "non-toxic," "0 chemicals," "0 pesticides." Or content expressing the product has low usage risk, such as "safe," "high safety," "few side effects."
- Z. Other: Select this option when the given information is not clearly related to any of the above options.

# Important Notes
1. Cautious Judgment: Incorrect identification may worsen a patient's condition or lead to other serious consequences. Please judge carefully and meticulously.
2. Authoritative Citations: If the text explicitly cites authoritative sources such as Baidu Baike, Sogou Baike, Wikipedia, book content, or book titles, judge it as "irrelevant" and return Z. Other.
3. Popular Science Content: If the text is purely for popular science purposes (e.g., from books, movies, etc.) and does not mention specific product names, judge it as "irrelevant" and return Z. Other.
4. Description of Patient Populations: If the text only describes a population of patients (e.g., books or events about AIDS patients or heart disease patients), judge it as "irrelevant" and return Z. Other.
5. General Therapeutic Effects: The text may contain words related to general therapeutic effects. These effect-describing words are mainly used to describe changes in function or state, highlighting the positive effects. They can be divided into several categories: Upward-trending words such as "save," "improve," "increase," "enhance," "strengthen," "boost," etc., emphasize the improvement and optimization of a certain ability or state. Downward-trending words such as "anti-," "inhibit," "lower," "suppress," "alleviate," "control," etc., focus on reducing, inhibiting, or controlling adverse factors. Elimination words such as "say goodbye to," "eradicate," "disappear," "completely," "cure," "get rid of," "solve," etc., emphasize the complete elimination and removal of problems or obstacles. Dosage-related words such as "only need," "just," "simply requires," etc., highlight ease of use and low cost. In addition, other words like "effective rate," "cure rate," etc., emphasize high efficiency and success rates. The common feature of these words is that they convey significant effects and advantages through specific verbs and adjectives.

Figure 7: The prompt of Health products in Single violation.

## D.2 TEMPLATE RELATED PROMPT

For template generation, data augmentation, and template-based inference, we present a series of prompts, each provided in its original Chinese and a translated English version.

---

**Seed Template Generation-zh**

# 任务描述
你是一名优秀的解题模版制定专家，专门设计用于内容风控的分析模板。你需要基于给定的管控规则，创建多样化的分析模板来识别违规内容。请仔细阅读给定的管控规则——这个规则定义了某一个文本/图片是否存在违规、以及具体违反了给定规则中的哪几条——逐步分析将给定规则转化为多个不同的待填充信息的解题模版。核心要求：1. 生成 10 个 独特的分析模板。2. 每个模板必须在逻辑流程、步骤划分或分析视角上完全不同。3. 严禁简单重复、换词改写或套用相似结构。4. 每个模板都应该有独特的分析角度和解题思路。5. 模板应该具有实用性，能够有效识别违规内容。
# 给定管控规则
## 管控类型：{TASK_RULE}
## 注意事项：{NOTICES}
# 参考例子
你需要生成 10 个模板。每个模板在逻辑流程、步骤划分或分析视角上必须是唯一的，严禁简单重复或换词改写。
## 可参考的模版步骤
1. 信息理解与关键词提取： 首先，仔细阅读并理解给定信息的核心内容，提取与规则描述相关的关键词。2. 逐项类别匹配： 其次，将提取的关键词和信息上下文，与每一个管控类型定义进行比对，记录所有初步符合的类别。3. 特殊条件判断： 接着，针对需要"通用治疗效果"的特定类别（如I类至L类），检查是否同时满足疾病/状况描述和通用治疗效果词的出现。4. "注意事项"审查： 然后，严格按照"注意事项"中的规则（如权威引用、科普内容、患病人群描述等）进行过滤，排除或确认Z选项。5. 最终结论形成： 综合以上匹配和审查结果，确定最终的一个或多个管控类型选项。如果没有任何匹配且不符合Z类特殊情况，则再次审视是否应为Z。6. 结构化输出： 最后，按照指定的JSON格式，清晰地阐述分析过程、列出识别的关键信息（关键词）和得出的结论。
# 输出格式
```json
[{"template_name": "text-template1", "steps": ["1、首先我们...", "2、其次可以...", ...]}, ... ]
```

---

**Seed Template Generation-en**

# Task Description
You are an expert in creating problem-solving templates, specifically designed for content moderation analysis. Based on the provided moderation rules, you need to create a diverse set of analysis templates to identify violating content. Please carefully read the given moderation rules—which define whether a piece of text/image has violations and which specific rules it violates—and progressively break them down into multiple, distinct, fill-in-the-blank analysis templates. Core Requirements: 1. Generate 10 unique analysis templates. 2. Each template must be completely different in its logical flow, step-by-step breakdown, or analytical perspective. 3. Simple repetitions, rephrasing, or using similar structures is strictly prohibited. 4. Each template should have a unique analytical angle and problem-solving approach. 5. The templates must be practical and effective for identifying violating content.
# Given Moderation Rules
## Moderation Category: {TASK_RULE}
## Special Considerations: {NOTICES}
# Reference Example
You need to generate 10 templates. Each template must be unique in its logical flow, step-by-step breakdown, or analytical perspective. Simple repetition or rephrasing is strictly forbidden.
## Sample Template Steps
1. Content Comprehension and Keyword Extraction: First, carefully read and understand the core content of the given information, extracting keywords relevant to the rule descriptions. 2. Category-by-Category Matching: Next, compare the extracted keywords and content context against each moderation category definition, and record all initially matching categories. 3. Special Condition Assessment: Then, for specific categories requiring "general therapeutic claims" (e.g., Categories I to L), check if both the description of a disease/condition and words implying general therapeutic effects are present simultaneously. 4. "Special Considerations" Review: Afterwards, strictly apply the rules from the "Special Considerations" section (such as authoritative citations, educational content, descriptions of patient populations, etc.) to filter, exclude, or confirm the Z option. 5. Final Conclusion Formulation: Synthesize the results from the matching and review steps to determine the final one or more moderation category options. If there are no matches and the case does not fit any special conditions for Z, re-evaluate if it should be categorized as Z. 6. Structured Output: Finally, follow the specified JSON format to clearly articulate the analysis process, list the key identified information (keywords), and state the conclusion.
# Output Format
```json
[{"template_name": "text-template1", "steps": ["1. First, we...", "2. Next, we can...", ...]}, ... ]
```

---

Figure 8: The initial prompt for generating a rule-based template, showing the original Chinese version (top) and its English translation (bottom).

---

**Data Augmentation (Template Extension) - ZH**

# 任务描述
你是一名优秀的解题模版制定专家，专门设计用于内容风控的分析模板。你需要基于给定的管控规则，创建多样化的分析模板来识别违规内容。现在我希望你对我给定的template进行不同风格的续写，尝试从新的角度和方法来解决问题。

# 续写要求
1. 生成 X个 独特的分析模板 2. 每个模板必须在逻辑流程、步骤划分或分析视角上完全不同 3. 严禁简单重复、换词改写或套用相似结构 4. 每个模板都应该有独特的分析角度和解题思路 5. 模板应该具有实用性，能够有效识别违规内容

# 给定管控规则
## 管控类型：{TASK_RULE}
## 注意事项：{NOTICES}

# 续写模板
{TEMPLATE}

# 输出格式
```json
[{"template_name": "text-template1", "steps": ["1. First, we...", "2. Next, we can...", ...]}, ... ]
```

---

**Data Augmentation (Template Extension) - EN**

# Task Description
You are an expert in creating problem-solving templates, specifically designed for content moderation analysis. Based on the provided moderation rules, you need to create a diverse set of analysis templates to identify violating content. Now, I want you to generate continuations for the given template in different styles, approaching the problem from new perspectives and with new methodologies.

# Continuation Requirements
1. Generate X unique analysis templates. 2. Each template must be completely different in its logical flow, step-by-step breakdown, or analytical perspective. 3. Simple repetitions, rephrasing, or using similar structures is strictly prohibited. 4. Each template should have a unique analytical angle and problem-solving approach. 5. The templates must be practical and effective for identifying violating content.

# Given Moderation Rules
## Moderation Category: {TASK\_RULE}
## Special Considerations: {NOTICES}

# Template for Continuation
{TEMPLATE}

# Output Format
```json
[{"template_name": "text-template1", "steps": ["1. First, we...", "2. Next, we can...", ...]}, ... ]
```

Figure 9: Template continuation: This prompt demonstrates how the initial template is extended and refined to accommodate more rules or details.

---

**Data Augmentation (Template Style Transfer) - ZH**

**# 任务描述**
你是一名优秀的解题模版制定专家，专门设计用于内容风控的分析模板。你需要基于给定的管控规则，创建多样化的分析模板来识别违规内容。现在我希望你对我给定的template进行不同风格的变换，尝试从新的角度和方法来解决问题，尝试从新的角度和方法来解决问题。

**# 续写要求**
1. 生成 X个独特的分析模板 2. 每个模板必须在逻辑流程、步骤划分或分析视角上完全不同 3. 严禁简单重复、换词改写或套用相似结构 4. 每个模板都应该有独特的分析角度和解题思路 5. 模板应该具有实用性，能够有效识别违规内容

**# 给定管控规则**
## 管控类型：{TASK_RULE}
## 注意事项：{NOTICES}

**# 续写模板**
{TEMPLATE}

**# 输出格式**
```json
[{"template_name": "text-template1", "steps": ["1. First, we...", "2. Next, we can...", ...]}, ... ]
```

---

**Data Augmentation (Template Style Transfer) - EN**

**# Task Description**
You are an expert in creating problem-solving templates, specifically designed for content moderation analysis. Based on the provided moderation rules, you need to create a diverse set of analysis templates to identify violating content. Now, I want you to create different stylistic transformations of the given template, approaching the problem from new angles and with new methodologies.

**# Transformation Requirements**
1. Generate X unique analysis templates. 2. Each template must be completely different in its logical flow, step-by-step breakdown, or analytical perspective. 3. Simple repetitions, rephrasing, or using similar structures is strictly prohibited. 4. Each template should have a unique analytical angle and problem-solving approach. 5. The templates must be practical and effective for identifying violating content.

**# Given Moderation Rules**
## Moderation Category: {TASK_RULE}
## Special Considerations: {NOTICES}

**# Base Template to Transform**
{TEMPLATE}

**# Output Format**
```json
[{"template_name": "text-template1", "steps": ["1. First, we...", "2. Next, we can...", ...]}, ... ]
```

---

Figure 10: Template style transfer: This prompt showcases the process of adapting the template's style to meet specific requirements or contexts.

---

**Identify the placeholders in the template - ZH**

# 角色与目标
你是一个专门优化指令模板的AI助手。你的任务是将一个"静态的"指令模板，转换为一个包含占位符的、可动态实例化的"元模板"。

# 核心概念
"元模板"保留了原始模板的逻辑步骤框架，但将其中描述具体实体（如"某个疾病"、"某种症状"）的通用词汇，替换为标准化的占位符（如 `[占位符名称]`）。这样做是为了让模型在处理具体任务时，能先用实际文本中的信息填充这些占位符，生成一个为该任务量身定制的、高度具体的执行指南。

# 转换规则
1. 保持结构不变：严格保留原始模板的步骤数量、顺序和核心逻辑。不要添加或删除步骤 2. 识别可变内容：仔细阅读每个步骤，找出那些在实际应用中会被具体信息替换掉的通用名词或短语。例如，"疾病"、"症状"、"治疗方法"、"产品"、"宣称"等词 3. 替换为标准占位符：将识别出的可变内容，用一个更通用的占位符，用于指代需要特别关注的、来自当前文本的任何特定信息。进行替换 4. 保持语言流畅：替换后，确保整个句子的表述依然通顺、自然。

# 优秀范例
这是一个完美的转换示例，请严格遵循这种转换模式。
Input: {Original_Template}
Output: {Meta_Template}

# 需要转化的模板
```json
{NEED_TO_CONVERT_TEMPLATE}
```

---

**Identify the placeholders in the template - EN**

# Role and Goal
You are an AI assistant specializing in optimizing instruction templates. Your task is to convert a "static" instruction template into a "meta-template" that contains placeholders and can be dynamically instantiated.

# Core Concept
"Meta-templates" retain the logical step-by-step framework of the original template, but replace general terms describing specific entities (e.g., "a certain disease," "a certain symptom") with standardized placeholders (e.g., `[Placeholder_Name]`). The purpose of this is to allow the model, when handling a specific task, to first fill these placeholders with information from the actual text, thereby generating a highly specific, tailor-made execution guide for that task.

# Conversion Rules
1. Maintain Structure: Strictly preserve the number of steps, their order, and the core logic of the original template. Do not add or delete steps. 2. Identify Variable Content: Carefully read each step to find generic nouns or phrases that would be replaced by specific information in a real-world application. Examples include words like "disease," "symptom," "treatment method," "product," or "claim." 3. Replace with Standard Placeholders: Replace the identified variable content with a standardized placeholder. This placeholder should be used to denote any specific information from the current text that requires special attention. 4. Ensure Fluency: After replacement, ensure that the entire sentence remains fluent and natural-sounding.

# Excellent Example
This is a perfect conversion example. Please strictly follow this conversion pattern.
Input: {Original_Template}
Output: {Meta_Template}

# Template to be Converted
```json
{NEED_TO_CONVERT_TEMPLATE}
```

Figure 11: Identifying critical placeholders in the template: This prompt highlights the extraction of key placeholders from the template to ensure flexibility and contextual alignment.

**Template-Based Reasoning Prompt - ZH**

# 任务描述
你是一个严谨的分析引擎。你的唯一任务是严格遵循下面提供的`推理模板`中的每一个步骤，对`给定信息`进行分析，并得出最终结论。模板中的每一步都是你必须执行的指令。

# 规则
{TASK_RULE}

# 给定信息
{QUESTION}

# 推理模板
{TEMPLATE}

# 推理流程
## 第一部分：整体分析
首先忽略细节，通读给定信息，形成一个初步的、整体的判断。
## 第二部分：搜集证据
在这一部分，请不要直接给出结论。而是针对模板中的 [占位符]（代表问题的关键、复杂、易错的方面）进行分析，来尝试证明你的整体分析中的结论。
1. 快速扫描：阅读用户提供的文本。
2. 提取核心实体：识别出文本中提到的具体疾病/状况、核心症状、以及提及的产品/疗法等。
3. 找到模板中的 [占位符]。
4. 对[占位符]进行初步分析和判断。
## 第三部分：综合研判与最终结论
综合以上`模板化推理过程`中每一步的分析结果，进行最终的判断，决定是否符合整体判断，还是根据第二部分的证据发现了新的结论。在你的结论中，必须：
1. 明确给出一个或多个最终的管控类型（A-Z）。
2. 详细阐述你的推理过程，说明你是如何整合各步骤的发现，最终得出这个结论的。

# 输出要求
你的唯一任务是判断文本是否触发了管控规则。请像一个精准的规则引擎一样工作，聚焦于寻找并匹配违规信号。
1. 第一部分：模板化推理过程，严格按照模板步骤，逐条展示分析过程和发现。
2. 第二部分：综合研判与最终结论，基于第一部分的分析进行汇总和阐述。
3. 最后用`\box{{}}`输出最终答案，格式为`\box{{A~Z中的一个或多个选项}}`，box内只能包含答案选项，不允许有其他任何文字。
## 示例
分析：... 结论是\\box{{A}}

---

**Template-Based Reasoning Prompt - EN**

**# Task Description**
You are a rigorous analysis engine. Your sole task is to strictly follow every step in the `Reasoning Template` provided below to analyze the `Given Information` and arrive at a final conclusion. Every step in the template is a command you must execute.

**# Rules**
{TASK_RULE}

**# Given Information**
{QUESTION}

**# Reasoning Template**
{TEMPLATE}

**# Reasoning Process**
## Part 1: Holistic Analysis
First, ignore the details and read through the given information to form an initial, holistic judgment.
## Part 2: Evidence Gathering
In this part, do not state a direct conclusion. Instead, focus on analyzing the [Placeholders] from the template (which represent the key, complex, and error-prone aspects of the problem) to try and substantiate the conclusion from your holistic analysis.
1. Quick Scan: Read the user-provided text.
2. Extract Core Entities: Identify specific diseases/conditions, key symptoms, and any mentioned products/therapies in the text.
3. Locate the [Placeholders] in the template.
4. Conduct a preliminary analysis and judgment for each [Placeholder].
## Part 3: Comprehensive Judgment and Final Conclusion
Synthesize the analysis results from each step of the `Template-based Reasoning Process` above to make a final judgment. Decide whether this confirms your initial holistic judgment or if the evidence from Part 2 has led to a new conclusion. In your conclusion, you must:
1. Clearly state one or more final moderation categories (A-Z).
2. Detail your reasoning process, explaining how you integrated the findings from each step to reach this conclusion.

**# Output Requirements**
Your sole task is to determine if the text triggers the moderation rules. Please operate like a precise rule engine, focusing on finding and matching violation signals.
1. Part One: Template-based Reasoning Process. Strictly follow the template steps, presenting the analysis process and findings for each item.
2. Part Two: Comprehensive Judgment and Final Conclusion. Summarize and elaborate based on the analysis from Part One.
3. Finally, output the final answer using `\box{{}}`. The format is `\box{{One or more options from A-Z}}`. The box must only contain the answer options and no other text.
## Example
Analysis: ... The conclusion is \box{{A}}

Figure 12: Template-based inference: This figure illustrates the process of applying the finalized template to perform context-aware reasoning or inference.

