# OpenReview forum: "Structuring Reasoning for Complex Rules Beyond Flat Representations"
_ICLR.cc/2026/Conference — ICLR 2026 Conference Withdrawn Submission_

### Official Review · Reviewer_YSjV · 2025-10-28

**Soundness:** 2
**Presentation:** 2
**Contribution:** 2
**Rating:** 4
**Confidence:** 4

**Summary:**

The manuscript introduces the Dynamic Adjudication Template (DAT), a prompting wrapper (qualitative analysis, evidence gathering, adjudication) that forces LLMs to reason over dense, inter-dependent rules instead of treating them as flat text.
A global–local template selector picks the best DAT from an automatically generated library. On the 16 k-sample e-commerce rule benchmark EVADE, DAT raises Qwen-2.5-7B partial accuracy from 34.1 % (CoT) to 62.5 % and outperforms GPT-4.1/DeepSeek-R1 CoT on five of six product categories.

**Strengths:**

This framework sees large, consistent accuracy gains on the dataset. Gains hold for ChatGLM, InternLM, Qwen-14 B (Table 1) and for VLMs Claude-3, GPT-4o, Qwen-VL-Max (Table 3).

Fig. 5 case study shows DAT explicitly extracts “minor exemption” clause missed by CoT, which is good for interpretability.

**Weaknesses:**

All experiments run only on EVADE (Chinese e-commerce ads); No evidence that templates transfer to English rules or substantially different ontologies.

Need to check hyper-parameter sensitivity and statistical significance.

No visualization of which template is picked for which input; selector could be trivially choosing the highest-accuracy single template.

Computational & cost overhead unreported - DAT adds three extra LLM calls per query; latency and token cost vs. CoT not tabulated.

**Questions:**

Evaluate DAT on one English statutory data set and one open-domain logic puzzle set; report macro-averaged gain.

Create an adversarial contradiction set, for example, generate 200 queries where two rules give opposite labels; report DAT precision and compare with CoT.

---

### Official Review · Reviewer_CRuu · 2025-10-30

**Soundness:** 2
**Presentation:** 1
**Contribution:** 2
**Rating:** 4
**Confidence:** 3

**Summary:**

This paper identifies that large language models (LLMs) struggle with complex, interdependent rule systems, as standard methods like Chain-of-Thought (CoT) treat rules as unstructured text and suffer from error propagation. To address this, the authors propose the Dynamic Adjudication Template (DAT), a three-stage framework inspired by human expert reasoning: (1) Qualitative Assessment for an initial judgment, (2) Evidence Gathering using dynamic templates with placeholders for targeted rule verification, and (3) Adjudication to synthesize the validated evidence into a final decision. The authors also introduce an automated pipeline for generating, filtering, and selecting these templates using a global and local (DPO-trained) selector. Experiments on the EVADE e-commerce benchmark show that DAT enables smaller LLMs to match or exceed the performance of larger models using standard CoT.

**Strengths:**

1. Clear Problem Definition: The paper addresses a well-defined and significant limitation of LLMs in applying complex, interdependent rules, which is a critical capability for domains like law, finance, and content moderation.

2. Intuitive Framework: The proposed three-stage DAT framework (Qualitative Assessment, Evidence Gathering, Adjudication) is logical, intuitive, and well-grounded in how human experts might systematically approach such problems, offering a more structured alternative to free-form reasoning.

3. Promising Efficiency: The central claim that a structured reasoning method can allow smaller, more efficient models to outperform larger, more general-purpose models (on specific tasks) is a valuable and compelling direction for research.

**Weaknesses:**

1. Limited Generalizability and Dataset Dependency: The entire DAT pipeline (template generation, filtering, DPO-trained selector, and evaluation) is developed and validated on a single, specific dataset (EVADE). It is highly questionable whether this complex system would generalize to other rule-intensive domains (e.g., legal text, medical guidelines) without a complete and costly re-generation and re-training process. The system risks overfitting to the specific structure and biases of this one benchmark.

2. High System Complexity: The proposed solution is far from simple. It introduces a complex pipeline that relies on multiple large models: a powerful generator (Gemini 2.5-Pro) to create templates and another model (Qwen-2.5-14B) to train the DPO-based local selector. This "scaffolding" complexity arguably offsets the final-use benefit of running inference on a smaller model (e.g., Qwen-2.5-7B). The paper does not provide a clear analysis of the total computational cost.

3. Weak Baseline Comparisons: The primary baseline is standard CoT prompting. While DAT shows clear improvements, CoT is a very general-purpose method. The comparison would be much stronger if it included other structured-reasoning baselines, such as verifier models, tool-use frameworks, or Retrieval-Augmented Generation (RAG) systems specifically designed to retrieve and apply rules.

4. Unconvincing VLM Results: The preliminary results on Vision-Language Models (VLMs) in Table 3 are mixed and undermine the paper's claims. For instance, DAT significantly degrades the performance of Qwen-VL-Max on the "Body" task (from 80.75 to 65.42). The paper's hypothesis that this is due to weak instruction-following in smaller VLMs does not explain this failure in a large VLM, suggesting the DAT approach may not be robustly applicable to multimodal reasoning.

**Questions:**

1. Could the authors elaborate on the required effort to adapt this system to a new domain, such as tax law? Would the entire template library and DPO selector need to be rebuilt from scratch?

2. What is the total computational cost of the DAT framework, including the one-time setup (generation, training) and the per-inference cost (selector + three-stage reasoning)? How does this compare to simply using a state-of-the-art model like GPT-4.1 with a more detailed multi-shot CoT prompt?

3. The performance seems to rely on "distilling" reasoning patterns from very large models (Gemini-Pro, Qwen-14B) into templates for a 7B model. How much of the performance lift is from the DAT method itself, versus this implicit knowledge distillation?

4. Can you explain the significant performance degradation for Qwen-VL-Max on the "Body" task? This seems to be a strong counter-example to the paper's core hypothesis.

5. How robust is the system to errors from the template selector? What happens if the global-local selector, $S_{final}$, chooses a suboptimal template for a given query?

---

### Official Review · Reviewer_7KCV · 2025-10-30

**Soundness:** 2
**Presentation:** 2
**Contribution:** 2
**Rating:** 4
**Confidence:** 4

**Summary:**

The paper proposes the Dynamic Adjudication Template (DAT), a three-stage prompting framework that forces an LLM to reason over dense, inter-dependent rule sets in a structured, verifiable way. DAT is automatically instantiated: a pipeline first generates hundreds of task-specific reasoning templates, filters them on a held-out set, and then trains a Global-Local selector (supervised + DPO) to pick the best template for every new query.

**Strengths:**

1)  The paper successfully identifies a clear and important gap in LLM reasoning for rule-based tasks.
2) Instead of hand-crafting prompts, the authors generate, score, prune and learn-to-select templates automatically.

**Weaknesses:**

1) All experiments are conducted on a single Chinese e-commerce moderation dataset. No transfer results are offered on legal statutes, privacy policies, financial regulations, or English data. Consequently, it is unclear whether DAT’s template distribution and selector remain effective when domain vocabulary, rule syntax or language change.
2) The authors only compare the method with CoT. Several advanced baselines are needed for comparison, e.g., ToT, GoT.
3) The DPO-trained Local Selector learns to prefer templates that historically produced correct final answers, but the reward signal is sparse (one bit per query). There is no guarantee that the chosen template generalises to unseen rule interactions, and the selector offers no post-hoc explanation for its choice. Interpretability is therefore limited to the execution trace, not to the selection process itself.
4) The ablation study focuses on the template selector's weighting and the necessity of the three reasoning stages. However, it lacks a critical component: ablating the template library itself. It is unclear how much of the performance gain comes from the structure of the DAT reasoning process versus simply having access to a large, diverse set of high-quality, pre-written "expert" prompts. An experiment comparing DAT against a static template (the single best template from the library) would help disentangle the contribution of dynamic selection from the quality of the templates.

**Questions:**

The construction of your DAT pipeline itself relies on state-of-the-art models like Gemini-2.5-Pro for template generation. This creates a dependency where boosting a small model requires access to a much larger one. Have you explored the sensitivity of your pipeline's performance to the choice of this "teacher" model? What is the minimum capability required for the model that generates the initial template library?

---

### Official Review · Reviewer_BbhW · 2025-10-31

**Soundness:** 2
**Presentation:** 3
**Contribution:** 2
**Rating:** 4
**Confidence:** 3

**Summary:**

This paper proposes a novel reasoning framework DAT for rule-intensive reasoning tasks, where LLMs struggle to interpret and apply numerous interdependent rules coherently. Unlike static prompting or unstructured reasoning, DAT introduces a template-based, adaptive mechanism that enforces disciplined progression through structured reasoning steps. The framework also includes an automated pipeline for template generation, filtering, and selection, enabling flexibility across varied rule systems. Empirical evaluations on complex rule-based e-commerce datasets show that DAT can improve reasoning accuracy and outperform larger models like Qwen-Max and DeepSeek-R1 on several subtasks.

**Strengths:**

The two limitations of current LLM reasoning identified by the paper are critical. The motivation is relevant and timely for domains such as law, e-commerce, and finance where structured rule interpretation is essential.

The proposed DAT introduces a three-stage structured process that reflects human reasoning. This decomposition provides an interpretable way to guide LLMs through rule-based reasoning tasks.

Across multiple models and datasets, DAT yields consistent and substantial improvements over CoT reasoning, making smaller models outperform larger LLMs significantly in some cases.

**Weaknesses:**

DAT’s stages are described procedurally but lack a formal logical or probabilistic grounding, making the method appear primarily heuristic. The connection between template-based reasoning and previous reasoning frameworks with Bayesian reasoning or symbolic reasoning is underexplored.

The experiments are limited on the e-commerce dataset, which represents only a narrow rule domain. I wonder if DAT can be applied to more general cases like visual-languange reasoning tasks.

While DAT improves accuracy with fewer parameters, the paper does not report time costs, token overhead, or template retrieval latency. The additional pipeline complexity could offset efficiency gains in real-world deployments. The discussion section does not provide qualitative examples of reasoning errors or limitations. A discussion of failure cases would make readers understand the border of DAT.

**Questions:**

Can the template library and selector generalize to domains with very different rule topologies, or must new libraries be generated for each domain? And are there mechanisms for compositional generalization when applying templates to unseen rule sets?

What are the computational trade-offs of DAT compared to CoT or ToT, especially in inference time or template retrieval overhead?

Could DAT leverage visual data during the Evidence Gathering phase, or are the templates still text-only?

---

### Note · Authors · 2025-11-13

I have read and agree with the venue's withdrawal policy on behalf of myself and my co-authors.